# Experimental Tests for Measuring Individual Attentional Characteristics in Songbirds

**DOI:** 10.3390/ani11082233

**Published:** 2021-07-29

**Authors:** Loïc Pougnault, Hugo Cousillas, Christine Heyraud, Ludwig Huber, Martine Hausberger, Laurence Henry

**Affiliations:** 1CNRS, EthoS (Éthologie Animale et Humaine)-UMR 6552, University Rennes, Normandie University, F-35000 Rennes, France; loic.pougnault@univ-rennes1.fr (L.P.); hugo.cousillas@univ-rennes1.fr (H.C.); christine.heyraud@univ-rennes1.fr (C.H.); martine.hausberger@univ-rennes1.fr (M.H.); 2Comparative Cognition, Messerli Research Institute, University of Veterinary Medicine Vienna, 1210 Vienna, Austria; Ludwig.Huber@vetmeduni.ac.at

**Keywords:** spontaneous attention, social and non-social visual attention, auditory attention, experimental tests, starlings, *Sturnus vulgaris*

## Abstract

**Simple Summary:**

Attention is at the core of all cognitive processes such as learning, memorization, and categorization. However, quantifying animals’ attention is challenging, and there is still a need to have standardized and easy-to-use tests. This article describes experimental tests aimed to evaluate spontaneous attention of a songbird, the European starling, in response to the presentation of different types of stimuli: visual non-social or social stimuli and auditory hetero or species-specific stimuli. Our results reveal intra-individual variations but also consistency of individual attentional characteristics in the presence of a species-specific stimulus or different auditory stimuli. They demonstrate further that attention is not a uniform concept and may depend upon the modality and the type of stimulus perceived.

**Abstract:**

Attention is defined as the ability to process selectively one aspect of the environment over others and is at the core of all cognitive processes such as learning, memorization, and categorization. Thus, evaluating and comparing attentional characteristics between individuals and according to situations is an important aspect of cognitive studies. Recent studies showed the interest of analyzing spontaneous attention in standardized situations, but data are still scarce, especially for songbirds. The present study adapted three tests of attention (towards visual non-social, visual social, and auditory stimuli) as tools for future comparative research in the European starling (*Sturnus vulgaris*), a species that is well known to present individual variations in social learning or engagement. Our results reveal that attentional characteristics (glances versus gazes) vary according to the stimulus broadcasted: more gazes towards unusual visual stimuli and species-specific auditory stimuli and more glances towards species-specific visual stimuli and hetero-specific auditory stimuli. This study revealing individual variations shows that these tests constitute a very useful and easy-to-use tool for evaluating spontaneous individual attentional characteristics and their modulation by a variety of factors. Our results also indicate that attentional skills are not a uniform concept and depend upon the modality and the stimulus type.

## 1. Introduction

Attention is defined as the ability to process selectively one aspect of the environment over others [1]. It involves several processes by which the nervous system apprehends and organizes sensory inputs and generates coordinated behavior [2]. Attentional mechanisms are essential to most cognitive processes, such as learning [3], memorization [4,5], and categorization [6]. Attentional processes allow individuals to adapt to their physical and social environments [7,8,9].

Most studies of animals’ attentional characteristics aimed to promote valid animal models of human attention disorders [10] and to explore the attentional mechanisms, the cerebral substrates which underlie these functions [7,11], as well as the phylogeny of these complex behaviors [8]. Tests used vary according to the attentional process being investigated, but they are generally based on operant conditioning [2]. The test evaluating attention used the most frequently is probably the “Five Choice Serial Reaction Time Task” (5-CSRTT) developed by Carli et al. [12] for rodents to investigate the main types of attention. During this test, a rat is placed facing five openings set in a horizontal array along a curved wall of the test chamber. A discriminative visual stimulus is presented, consisting of a brief illumination of one of the five openings (0.5 s). When the subject reaches the opening with the stimulus, it receives a positive reinforcement such as a food pellet. This task tests the ability of rats to divide attention among a number of locations, to sustain attention over a large number of trials (about 100), and to express selective attention by ignoring brief bursts of white noise while detecting the visual targets [10]. Recently, Rochais et al. [13] successfully adapted the 5-CSRTT for horses. However, this test requires extensive training, taking weeks or even months to learn the task according to the species [14], and behavior may be modulated by inter-individual differences related to individuals’ motivation to search for food. On the contrary, by analyzing unconditioned behavior, animals’ attention can be recorded rapidly and in an economical way [2]. Analyses of the spontaneous expression of attention reveal variations related to intrinsic (e.g., age, sex, [15]) or extrinsic (e.g., human–animal relationship, [16,17] factors). Few studies, however, measured the structure of spontaneous attention. Blois-Heulin [18] showed that the structure of social attention varies according to social organization and distinguished short glances from more durable gazes (i.e., longer attention, lasting more than 1 s). Various studies based on social or human-animal contexts suggest that visual attentional structures present intrinsic differences between species [8,19,20].

Characterizing and evaluating attention and its structure in order to assess individual variations and their causes in a predictable way is a challenge. Most studies of spontaneous attention were performed either in naturalistic settings or in tests based on social visual stimuli (e.g., [21]).

Songbirds are relatively little studied in this respect. Attention is, however, one of the mechanisms proposed to explain selective song copying [22,23]. This is especially the case of European starlings *(Sturnus vulgaris)* for which social selective attention was proposed as a major factor involved in song copying between conspecifics [24,25,26,27]. The aim of the present study was to test the feasibility and the interest of developing attention tests that could be used in further studies to examine the impact of different factors, experience in particular, on individual attentional characteristics. We adapted the social visual attention tests (attention to conspecifics) developed by Range et al. [19] for keas (*Nestor notabilis*), dogs (*Canis familiaris*), and humans (*Homo sapiens*; pre-school children) and the non-social (visual or auditory) attention tests recently developed by Rochais et al. [13,28] for horses. Range et al. [19] demonstrated that the attentional behavior of the three species they studied depended on age/sex/identity of the model but that birds (keas) looked longer at a feeding than a non-feeding model, while children paid more attention to a model manipulating objects. Dalmaso et al. (2020) [9] indicated that gaze-mediated orienting of attention in humans can be shaped by social variables. Rochais et al. [28] found that individual variations of visual attentional characteristics remained relatively stable over time and across situations, as individuals ranked similarly after 6 months and the results of this test were predictive of attention in other contexts (e.g., working situation). They found opposite results between the visual and the auditory attention tests, the former predicting attention and the latter distractibility in working situations. In humans, it is well known that one can be more easily distracted from an ongoing task by auditory stimuli that do not require one to look than by visual stimuli [29].

The aim of the present study was to develop tests of attention as a tool for future comparative research on the impact of intrinsic or extrinsic factors on songbirds’ attentional processes. European starlings’ vocal repertoire and social integration present important individual variations, and song learning is highly dependent upon attention towards social models (e.g., [30,31]). Therefore, we expected attentional characteristics to present important individual variations and, in these song learners, visual and auditory attention to converge when submitted to species-specific stimuli. Three tests were used to characterize the structure of spontaneous attention of adult male starlings that did not have any specific training. Overall, starlings, as with most species, react to auditory stimuli by increasing their visual attention towards the sound source so that the same types of measures (e.g., number and duration of gazes towards the source) can be used to quantify attention towards visual or auditory stimuli (e.g., [32]). In accordance with Cohen [33], we differentiated attention capture/getting that triggers an orienting response [34] (and attention holding which is generally associated with a decrease of movement and more focused attention [24]. Passerine birds often scan the environment by repeated head movements associated with short looks (“glances” of less than one second) but may focus and remain immobile when gazing (e.g., [32]). Therefore, we measured attention characteristics on the basis of these two types of visual attention. A previous study showed that gazes towards a novel heterospecific stimulus are more lateralized than glances in starlings, suggesting further that they reflect individual characteristics and perception of different types of stimuli [35].

## 2. Material and Method

### 2.1. Subjects

The tests ran between 1 April 2017 and 14 April 2017 included 10 male starlings that were wild-caught as adults (they were more than 2 years old according to their throat feathers [36,37]) in Normandy, 4 individuals in November 2012 and 6 in November 2014. They were then kept with other starlings in mixed groups in outdoor aviaries with open shelters, grassy grounds, water and food ad libitum, and numerous perches. The choice of sample size was based on experimental logistic constraints but also and mainly on a wish to keep a balance between statistical validity and the “reduction” principle of the 3R ethical principles that aim to reduce as much as possible the number of experimental animals involved in research programs, especially as we used wild-caught animals.

Before the attention tests in April 2017, the individuals were transferred to the EthoS laboratory (UMR6552) on the Rennes1 University campus and placed in individual wire cages (60 × 39 × 65 cm), where they had an unlimited access to food and water throughout the course of the study. The cages had one grid side in the front and three opaque sides.

After 4 days of habituation without any stimulation, they were submitted individually to 3 experimental conditions with the goal to characterize their spontaneous attention towards 3 different types of stimuli.

### 2.2. Experimental TESTS

Three tests were conducted: (a) the non-social visual attention test (VAT) inspired by Rochais et al. [13], (b) the social visual attention (SVAT) inspired by Range et al. [19], and (c) the auditory test (AAT) inspired by Rochais et al. [28]. Both VAT and AAT were performed during the first week of experimentation, and the visual social test was performed during the second week. We chose to perform the non-social tests first, as we supposed that the social tests may have a more excitatory or motivational effect on the birds, which could then bias the results of the non-social tests (e.g., seeking more conspecific behind the screen or the loudspeaker). The order of the non-social tasks was changed between two subgroups of birds: five birds were tested first in the non-social visual task and then in the auditory task, and the other five birds were tested in the opposite order. During the second week, the 10 birds were tested in the social visual test.

### 2.3. The Non-Social Visual Attention Test (VAT)

Each cage was located in an anechoic chamber (35 dB attenuation). An LCD screen 15” (1024 × 768 pixels, AccuSync LCD52VM©) was attached to one lateral opaque cage side. Two perches were positioned respectively 15 cm and 35 cm from the screen. A USB camera (Logitec CarlZeiss Tessar2.0/3.7©) was used to record the whole cage screen, and the cameras were controlled by an external computer (HP elitbook1470P©).

The stimulus was a green spot (2 cm in diameter) selected in accordance with the perceptual capacities of the birds [38]. Previous studies showed that European starlings can react to 2D images (e.g., [39,40,41]). Moreover, this novel moving visual stimulus was chosen to avoid biases due to familiarity and to take into account starlings’ abilities to detect movement [12]. It was projected on the black screen for 2 min when the bird was positioned on the perch nearest the screen and facing it. The trajectory of the spot followed the same pattern of movement as in Rochais et al.’s [13] VAT by using the Processing software©. The object was continuously moving, therefore, it slowly transformed into a successive pattern following a circular movement (Figure 1). As suggested by Davidson et al. (2014) [42], we adjusted the test duration to species-specific characteristics. Songbirds are known for presenting rapid shifts of attention [43], much more rapid than horses. Each bird was tested four times, once each half-day for two consecutive days. Recording began 2 min before the stimulation was switched on and continued for 2 min during the stimulation. Thus, in all, we obtained 16 min of recording for each bird.

### 2.4. The Social Visual Attention Test (SVAT)

For this test, five pairs of starlings were randomly constituted from the initial group of 10 subjects. Observations in their aviary revealed no clear hierarchical ranking, therefore, we could not take this parameter into account, although some authors found a relationship between hierarchical status and cognitive abilities in starlings [44] or between dominance and attention in other species [45].

Four days before a test, each dyad was placed in an experimental wire cage (121 × 50 × 39 cm) separated into two parts by a white opaque plexiglass board with one bird on each side of the cage. A horizontal slot (6 × 2 cm) at bird’s eye level was drilled in the board. The length of this horizontal slot could be modified using a white sliding plexiglass tongue. The behaviors of the test bird (named “A”) were recorded by a camera (Sony Handycam, DCR-SR75^®^) located 1 m in front of the experimental cage. To reduce the probability of interactions between the two subjects, they were deprived of food one hour before the experiment and the observed bird (hereafter, “B”) had access to palatable food (insectivorous mix for birds).

The test bird (“A”) had visual access to his conspecific (“B”) twice a day for 5 min: in the morning, with the slot fully opened (6 cm) so that bird “A” could learn the presence of his conspecific “B”; and in the afternoon, when the slot was only opened 1 cm to promote potential monocular observation of “B” by “A” (hence potentially laterality). We decided to systematically test our birds first in the morning with a large opening (6 cm) for giving the possibility to the tested bird (A) to be aware of its congener presence (B) and to see him binocularly. In the afternoon, the limited opening (1 cm) allowed only monocular attentional behaviors. Pilot experiments showed that the birds did not always see that there was a conspecific when the slot was small (see Range et al. [19], who allowed a familiarization period to birds before tests). Since we wanted to make sure the animals saw that they had access to a visible neighbor, we were obliged to have a fixed order, even though, of course, random allocation would have been ideal.

### 2.5. The Auditory Attention Test (AAT)

For this test, subjects were placed individually in the experimental cage without any visual contact with conspecifics. The 10 cages were placed against a wall, allowing simultaneous video recording of all birds. The auditory stimuli were broadcasted using a loudspeaker (Philips SPA5300 2.1^®^). A camera (Sony Handycam, DCR-SR75^®^) was placed above the loudspeaker (see [35]). The two devices were placed 2 m in front of the experimental cages, and the behaviors of all the subjects were recorded at the same time to avoid habituation due to repetition of the stimulus. Two types of stimuli were broadcasted: (a) 4 different whistles from unfamiliar individuals recorded in New Zealand to ensure that the subjects could not have heard them before and (b) 4 different vocalizations of humpback whales (*Megaptera novaeangliae*) as unknown heterospecific stimuli. Durations (0.50 s) and intensities (~70 dB at two meters) were standardized for all stimuli using ANA software [46]. Four different vocalizations (with different acoustic structures) of each type, one of each per half day, were broadcasted in random order and various time intervals (between 6 to 21 min) between stimulations to avoid biases due to habituation. Therefore, birds heard each stimulus only once. Video recording began 1 min before the stimulation and finished 1 min after the stimulation; in all, we obtained 8 min of recording for each type of stimulus.

### 2.6. Data and Statistical Analyses

Video recordings were analyzed using continuous focal sampling [47]. All analyses were made by the same single observer (LP), but interobserver reliability was tested with a second observer on a sample of recordings (Christiane Rössler), leading to 84% agreement, a reasonable assessment [48], especially on very short items such as glances.

The parameters measured were:Glances: monocular or binocular looks towards the stimulus lasting less than one second;Gazes: a monocular or binocular look towards the stimulus lasting more than one second;Total gaze duration: total time during which a subject gazed at the stimulus during a test session.

The distinction between gazes and glances was based on previous studies [20,49]. Furthermore, the first reaction after the stimulus presentation was also recorded (see also [41])

Normality and homogeneity of variance were assessed by inspection of residuals with Shapiro-Wilk W tests. Given the non-normality distribution of data, we ran non-parametric statistical tests [50] using R 3.3.2 software^®^ (R Core team, Vienna, Austria) [51] with a significance threshold set at 0.05. The aim of this work was to assess individual variations in attention structure for each given test separately using two aspects (number of glances and gazes, duration of gazes). Since the question and the data were quite simple and the sample size was reasonable, we stayed on “conventional” simple non-parametric statistical analyses. In the same line, we compared temporal changes in behavior within tests using Wilcoxon signed-rank test. Spearman’s correlations tests were used to assess possible individual consistency between tests (i.e., VAT; SVAT and AAT), since it compares the relative ranking of given individuals independently of the actual precise measures (Appendix A). In the figures, we did, however, use the rate per minute in order to facilitate an overview of the results at the study’s level.

## 3. Results

### 3.1. Visual Attention Tests

#### Non-SOCIAL (VAT)

All starlings paid attention to the stimulus, and none expressed any withdrawal or any other fear behavior. The first reaction of all of them when the stimulus appeared was to glance at the screen. However, frequencies of glances did not vary significantly between before and during the stimulation (Wilcoxon signed-rank test, T = 10, N = 15 *p* = 0.2). On the contrary, projection of the visual stimuli was associated with a significant increase of the frequency and the total duration of gazes (T_Gaze_ = 5, N = 10, *p* = 0.02; T_Total gaze duration_ = 1, *p* = 0.007), but not of glances (T = 15, N = 10, *p* = 0.2) towards the screen, although glances were overall more frequent than gazes at all times (T = 0, N = 10, *p* = 0.006) (Figure 2a). Starlings looked more at the stimulus with monocular than binocular gazes (T = 0, N = 9, *p* = 0.006) and glances (T = 1, N = 10, *p* = 0.007), but no laterality bias could be observed (*p* > 0.5 in all cases).

### 3.2. Social Visual Attention (sVAT)

The birds expressed more glances than gazes towards their conspecifics no matter the size of the slot opening (Wilcoxon signed-rank test, T_Max_ = 0, N = 10, *p* = 0.006; T_Min_ = 3.5, *p* = 0.01; 1, Figure 2b). Total gaze durations and frequencies of gazes did not differ significantly between minimal or maximal slot opening tests (T_gaze duration_ = 12, N = 9 *p* = 0.2, T_gazes frequencies_ = 3, N = 7, *p* = 0.06), but the subjects glanced more when the slot was large than when it was small (T_glances frequencies_ T = 2.5, N = 10, *p* = 0.01). Interestingly, when the slot was fully open, although they had the possibility to use either their binocular or monocular visual fields to look at their conspecifics, starlings performed more monocular than binocular glances and gazes (T_Glance_ = 8, N = 10, *p* = 0.04; T_Gaze_ = 8, p0.04). When using monocular vision, they still performed more glance than gazes (T = 0, N = 10, *p* = *0*.005). However, we could not evidence a laterality bias (preferred side) (T_Glance_ = 17.5, N = 10, *p* = 0.3; T_Gaze_ = 9.5, N = 9 *p* = 0.1; T_Total gaze duration_ = 4, N = 7, *p* = 0.09).

### 3.3. Auditory Test (AT)

When the auditory stimuli were broadcasted, all individuals displayed attentional behaviors (glances and gazes) in the direction of the loudspeaker. Although birds expressed more glances than gazes (Wilcoxon signed-rank test, T = 0, N = 10 *p* = 0.006 in both cases) (Figure 2c), their attentional structure differed according to the type of stimulus. Following the broadcast of heterospecific sounds, all subjects increased glances towards the loudspeaker (Wilcoxon signed-rank test, N = 10, T = 1, *p* = 0.006). Frequency or total duration of gazes did not change significantly (T_Gaze_ = 24, *p* = 0.7; T_Total gaze duration_ =23, *p* = 0.6). On the contrary, in response to the conspecific stimuli, all subjects increased numbers of gazes and total duration of gazes (T_Gaze_ = 0, N = 10, *p* = 0.006; T_Total gaze duration_ = 0, N = 10, *p* = 0.006), but numbers of glances did not change significantly (T = 11, N = 10, *p* = 0.09). Overall, our subjects emitted more glances (total number) towards the loudspeaker in response to the heterospecific than to the conspecific stimuli (Wilcoxon signed-rank test, N = 10, T = 3, *p* = 0.01) and more gazes and longer total gaze durations in response to the conspecific than the heterospecific stimulus (N = 10, T_Gaze_ = 0, *p* = 0.006; T_Total gaze duration_ = 0, *p* = 0.006) (ratios), Figure 2c).

### 3.4. Individual Consistency between Tests

The coefficients of variation observed for the different traits in the different tests showed that individuals differed most in gaze frequency and/or duration in the presence of the visual social stimulus and the heterospecific auditory stimulus. Interestingly, the numbers of glances towards their conspecific during the visual social test (minimal opening) and the numbers of glances after the broadcast of a conspecific vocalization were positively correlated (Spearman correlation test, rs = 0.70, N = 10, *p* = 0.05, Figure 3a).

Thus, the individuals the most attentive to the view of a conspecific were also the most attentive to conspecific sounds, but only in terms of repeated glances. There was no evidence for any other correlation of attentional data between the different tests, indicating no consistent attentional individual stability of their reactions towards visual versus auditory non-social stimuli or towards social versus non-social visual stimuli (Spearman’s correlation, for all *p* > 0.05). However, in the auditory test, total gaze durations towards the loudspeaker after the broadcast of a conspecific song and after a hetero-specific song were positively correlated (Spearman’s correlation, rho = 0.79, *p* = 0.01; Figure 3b).

## 4. Discussion

The results of the present study demonstrate the feasibility and the importance of attentional tests adapted for European starlings for the investigation of spontaneous attentional structures. These structures show individual variations and depend on the type of stimuli presented. Overall, the starlings’ attentional structure differed according to the target of their visual attention. They showed more durable attention when looking at an unusual non-social object than when they had access to the view of a familiar live conspecific. In all cases, they preferred to use a monocular visual field although without any laterality bias. Results reveal that starlings pay more attention towards unusual visual stimuli and conspecific auditory stimuli, whereas they glanced more at social visual stimuli and heterospecific auditory stimuli. These results suggest opposite patterns between visual and auditory social attention comparable to Rochais et al.’s [28] observations showing opposite individual processes between these two modalities. Nevertheless, the numbers of glances towards visual and auditory species-specific were correlated, suggesting that social motivation could trigger bimodal attention. Although we found no correlations between social and non-social visual tests, individual responses to the two types of auditory stimuli (conspecific and heterospecific sounds) were correlated, suggesting that individual auditory attention was more consistent. Individual variations were the highest for gazes, i.e., for more durable attention.

Classical experimental attention tasks require individuals to monitor the location of a stimulus to obtain food and are based on operant conditioning. Thus, differences between individuals then are related to an individual’s motivation to work to obtain food, and reward delivery can alter the processing of specific stimuli directly by increasing their attentional priority [10,12,53]. Moreover, training in operant conditioning can take time (e.g., 5-CSRTT: 2–4 months for rats [54]; 3–5 months for mice [14]. In addition, long-term experimental procedures can negatively affect a subject’s attentional function, as it may experience repeated stress such as restriction of food or water [55]. Therefore, these procedures are not adapted to study the “spontaneous attention” of a large number of animals. The three attentional tests we applied (i.e., VAT, SVAT, AT) for starlings and adapted from pre-existing attentional tests [13,19] proved again useful for assessing individual differences in spontaneous attention and variations in attentional structure according to the type of stimulus and the modality involved.

All our subjects performed many glances, and this could be related to starlings’ status of “prey-species”, which requires paying permanent attention to the environment while performing other behaviors such as feeding and to change rapidly the direction of visual attention rather than to fix their attention during a long period [7].

The attentional structure of our starlings appeared to depend both on the targets and the modality involved. As Davidson et al. (2014) [42] pointed out, gaze behaviors must be described in term of “observable behavior”. For example, a shorter latency and a longer time of looking may reflect gaze preference. Our birds paid more sustained attention, through an increase of gaze occurrences and their total duration, to unknown auditory or visual stimuli. However, glances were expressed more towards visual or auditory stimuli that they already knew, such as a conspecific or a “neutral” stimulus. These results suggest further that longer and frequent gazes reflect a higher level of attention than repeated glances (see also [21]).

Studies showed that, when facing a usual stimulus, carrying no information of interest, horses quickly scanned their environment by glances and then resumed their previous activities [56]. Conversely, Rochais et al. [13] described fixed attentional behaviors towards an unusual visual stimulus, revealing selective attention promoted by endogenous attentional processes. In the same context, our birds demonstrated a similar visual attention pattern. This same kind of behavior can be noted in a natural environment during foraging. Indeed, detection of food can be difficult (i.e., cryptic items such as seeds or insects) so that an individual must base its choice on particular aspects of the visual target [57,58,59]. Goto et al. [60] demonstrated a similar type of spontaneous visual attention in blue jays (*Cyanocitta cristata*) using the “search image” paradigm proposed by Tinbergen [59]. An important point is that, contrarily to Rochais et al. [13], we were able to remove the bias potentially induced by the presence of a human experimenter during the attentional test by using an electronic apparatus. This new method would be easy to use and to adapt the VAT test for other species in this same way to estimate spontaneous non-social visual attention.

Playback of species-specific vocalizations (i.e., AAT) induced starlings’ sustained attention. Poirier et al. [24] observed a global reduction of the activity of young starlings raised without adults when adult songs were broadcasted, suggesting an increase of auditory attention. These observations were confirmed by Ten Cate [23]: young zebra finches (*Taeniopygia guttata*) similarly approached and looked at an adult that was vocalizing. Because they lack external auditory organs, we cannot conclude which modality was privileged in this context. However, the head movements observed (see also [32]) suggest that the attentional process of young birds towards an adult was bimodal, i.e., auditory and visual attention. George et al.’ studies [39,61] revealed that the responses of auditory neurons in the primary auditory area of starlings was influenced by the broadcast of social visual stimuli. This bimodal treatment of information allows individuals to perceive information concerning the emission source of the stimulus [62]. Thus, during playback of conspecific auditory stimuli, starlings probably tried to complete the auditory information with visual information, orienting their gazes towards the emission source. On the contrary, our subjects expressed a majority of glances after hearing the hetero-specific auditory vocalization, which, according to San Miguel et al. [34], may have captured attention and triggered an orienting response but also induced “curiosity”, i.e., visual exploration [35]. Yet, total gaze durations appeared to remain consistent for each individual between these two auditory attentional tests, indicating interest differences between conspecific versus hetero-specific stimuli rather than differences between preferred modalities.

In addition, visual social attention was expressed mainly by glances in our experimental situations. Social life requires one to be attentive towards others and their actions [21]. Attention—particularly social attention—is modulated by social factors in most social species, including humans (Dalmaso et al. [9,45]. One important aspect is the quality of social relations between individuals such as hierarchical rank [63] or affinities ([30]. Blois-Heulin [18] and Lemasson et al. [64] showed opposite trends of visual attention between red-capped mangabeys (*Cercocebus torquatus torquatus*) characterized by a “despotic” type of society, where group members focus their attention towards the dominant male, and Campbell monkeys, characterized by a tolerant type of society where members’ visual attention network reflects affinities. In rhesus macaque (*Macaca Rhesus*), the lower ranking individuals reacted quicker than high ranking animals to visual cues from conspecifics [63]. Studies performed in large naturalistic aviaries showed that most social relationships were expressed through tolerance and spatial proximities between preferred partners, while the hierarchy was more rarely expressed and only in some particular contexts [44,65,66]. This was also the case in the observations performed on our experimental birds when in groups (Pougnault et al. in prep). In canaries, observational learning was enhanced when watching a preferred social partner [67]. We did not investigate the role of social status nor of being or not being social partners on the attentional pattern. Further studies will have to be developed in order to test specifically this question. At least, these standardized tests can help testing further such questions. Finally, there are differences between species in social attention due to other reasons than just social. Starlings express attention towards conspecifics while feeding in the form of “scans” (i.e., glances) or gazes if obstacles are present (e.g., tall grass), which allow them to anticipate a potential predation risk but also to locate a food source by looking at the body position of their conspecifics [68,69,70]. This is different from what was found in keas, which paid quite long attention to a conspecific that was feeding (up to 50% of the total time of demonstration) [19]. Jackdaws are much less inclined to watch conspecifics than ravens [21]. Finally, in our tests, birds emitted a majority of monocular visual behaviors, even when they had the possibility to look at their conspecific with binocular vision. Given the width of the opening (i.e., 2 cm width), their beak may have constituted a handicap because it could obstruct their field of view, and this could explain this result [71].

Individual differences clearly appeared, showing consistency mostly for auditory stimuli but also a transmodal consistency for social stimuli. Limited vision through a narrow opening or lack of vision of the conspecific seemed to induce similar individual differences in visual and in auditory attention, and this probably reflects individual differences in the motivation to seek social contact [40]. Future studies of larger samples with various social experiences should enlighten the processes underlying such individual differences and show how social attention and social motivation are interrelated (Perret et al. in rev., Henry et al. sub.). Horses’ responses to the auditory test were influenced by their welfare state [72] and by the visual test by their working conditions (Rochais et al. sub.).

## 5. Conclusions

This study aimed to validate new methodological tools, inspired from those used for other species, to evaluate individual starlings’ attentional characteristics. Given our conclusive results, these methodologies offer interesting ways to assess birds’ spontaneous attention whilst avoiding extensive training that can imply weeks or even months of training.

Our three tests appear to be promising and simple tools to investigate and predict the attention characteristics of several species. This kind of investigations seems able to highlight intrinsic (e.g., age, welfare state) and/or extrinsic (e.g., living conditions) factors of variations in attentional structure.

## Figures and Tables

**Figure 1 animals-11-02233-f001:**
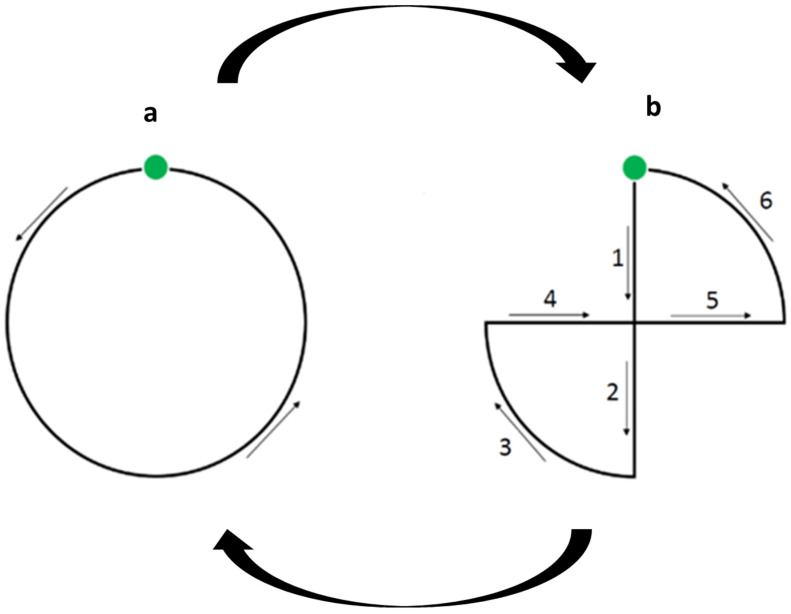
Visual stimulus used during the non-social visual attention test, adapted from [26]. The light was moved repeatedly, regularly, and continuously over 20 s using a computer program, starting anticlockwise (**a**) to continue over a “8” shape (**b**). Six cycles were broadcasted during each 2 min test.

**Figure 2 animals-11-02233-f002:**
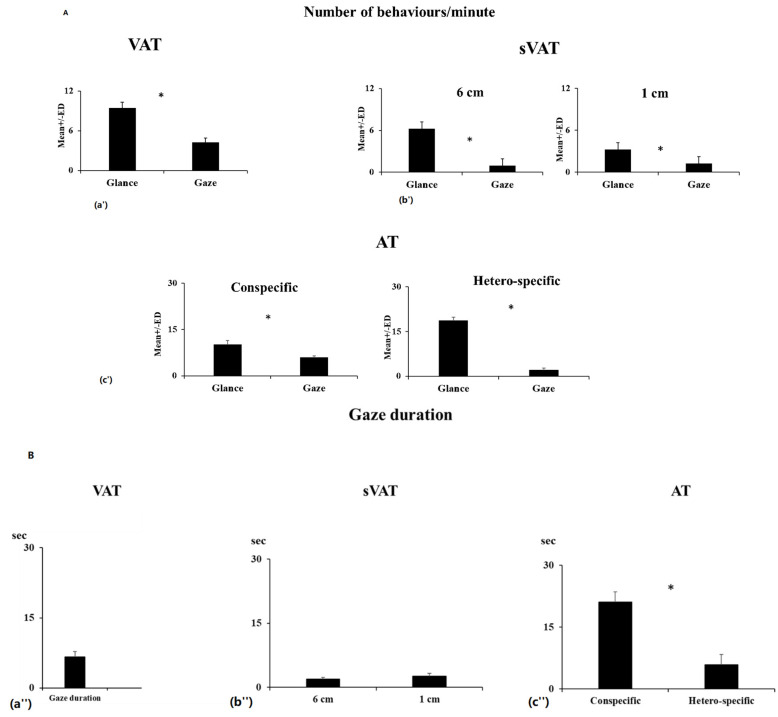
(**A**). Attention characteristics during tests: occurrences of glances and gazes, (**B**): duration of gazes during stimulus presentation, (data: mean number per minute ± ED [52], *: Wilcoxon test: *p* ≤ 0.05): (**a′**,**a″**) Non-social visual test (VAT). (**b′**,**b″**) Social visual test (sVAT) with maximum opening (6 cm) and minimum opening (1 cm). (**c′**,**c″**) Auditory test (AT): attention characteristics after the broadcast of conspecific vocalizations and hetero-specific (Whale).

**Figure 3 animals-11-02233-f003:**
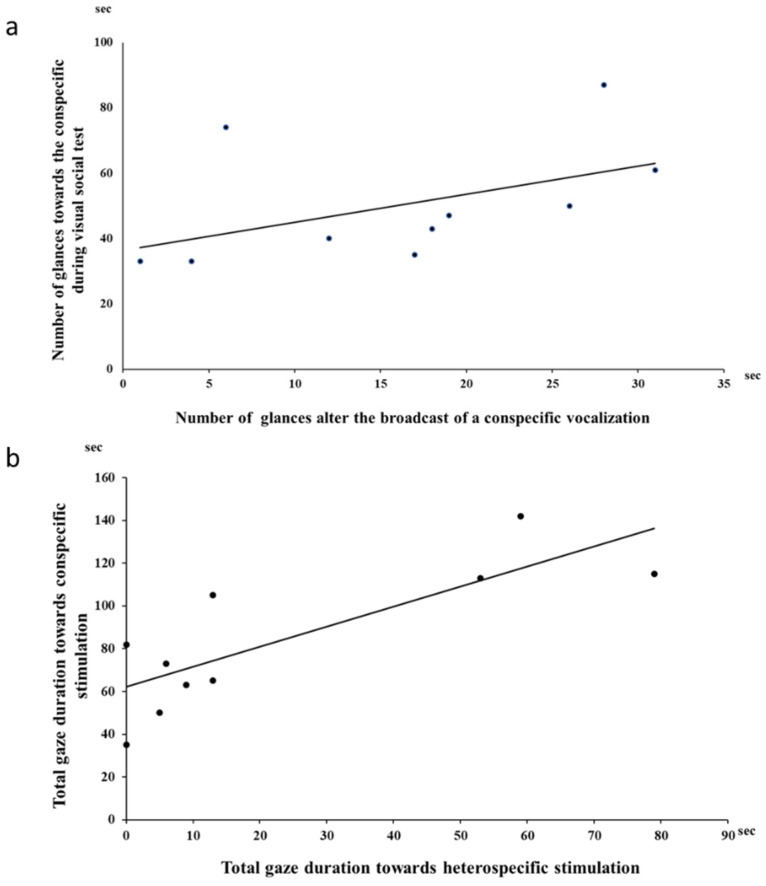
(**a**) Numbers of glances towards a conspecific during the visual social test (minimum opening) in relation to numbers of glances after the broadcast of a conspecific vocalization (correlation rs = 0.70, N = 10, *p* = 0.05). (**b**) Total gaze durations after the broadcast of the conspecific vocalization in relation to total gaze durations following the hetero-specific vocalization during auditory tests (correlation r = 0.79, N = 10, *p* = 0.01). Each dot represents an individual.

## Data Availability

Data are available on reasonable request to the corresponding author.

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
