# Peer review of "Experimental Tests for Measuring Individual Attentional Characteristics in Songbirds"

_animals, 2021, doi:10.3390/ani11082233_

Round 1
Reviewer 1 Report
In this study, the authors conducted three different experiments to assess spontaneous attention: non-social visual attention test (VAT), social visual attention test (SVAT) and auditory attention test (AAT). With conspecific and unfamiliar (heterospecific ) using adult European starlings (10 males individuals). The results showed that attention measured by scans and gaze vary between stimulus type and modalities.
The experiments adopted from the previously published papers and altered to suit the European starlings. Therefore, the methods are not entirely novel, and application to European starlings showed a potential to use in other songbirds. However, the experimental methods need to be clearly described, and several experimental faults need to be fixed before accepting the manuscript for publication.
Major comments:
- The comparisons made between tests with the number of gazes and the number of scans invalid due to time slot differences. For example, the authors did not use the same amount of stimulus exposure time for visual and non-visual tests for count comparisons (VAT 2 min per individual, SVAT 8 min per individual). The gaze or scan rate would have been the best measure when the time slots were different in each test. As it stands, authors compared duration for two or, in some cases, three tests with varying times of exposure using counts as the response variable, and these comparisons were not valid. Authors could have used mixed models with Poisson or Negative binomial distribution for count response data to control the individual difference and the exposure time as the offset value to account for the differences.
- In addition, the following problems exist in each experiment.
In the non-social visual object test (VAT), the time exposed the unusual object displayed on the screen not mentioned. The object presented with two different patterns (anti-clockwise as the figure indicated eight shapes). There was no information on how long each circulation pattern exposed to an individual within 2 min. I see authors refer to Rocha’s et al. 2017 for details, but again, the test was done for horses using different experimental setups, and the exposure time was different (5 min).
In the social-visual test (SVAT), the conspecifics were always exposed with 6 cm slot first and then 1cm opening; clearly, an order effect influenced the results. The proper way to experiment to eliminate the possible order effect is randomly alternating the slots ( 6 cm or 1cm) per individual.
In the auditory test (AAT), the conspecific and heterospecific calls were obtained from a single individual respectively (page 5, line193-198), and playbacks prepared with replicating them. This playback experiment, unfortunately, a pseudo- replicated one and may not provide a valid test. Please see: Kroodsma et al .2001 ( Animal Behaviour) DOI:10.1006/anbe.2000.1676
- Throughout the manuscript, I struggled to understand whether the authors used matched-pair design to compare individuals between tests, despite the stimulus exposure time difference. For example, they matched the same individual between tests and reported the result or used the cumulative measure for all the individuals and compared the measures. This needs to be clarified, specifically where the correlation analysis performed (e.g., figure 4 a) and whether each data point represents the same individual tested in both tests (conspecific visual social vs conspecific vocalization).
Overall, the presentation of data can further improve. First, include the missing graph 5; then graph 4-a, is invalid as the time slot differences in stimulus exposure time. Figure 3 b, c showed redundant information and present in table 1 and 2; therefore, authors may include two graphs: 1) the number of glances and glance duration; 2) the number of gazes and gaze duration. Figure 1 c appeared in another published paper, indicated that authors use the same photo for several publications.
Minor comments :
Page 1, line 31-32: The statement is invalid if the authors have used time interval for two tests; there is no information on how the green square move or the exposure time.
Page 1, line 33: The individual variation between tests were not established; please see the major comments above.
Davidson et al., 2014. Animal Behaviour 87, 3-15 should be cited to give readers different gaze definitions. https://doi.org/10.1016/j.anbehav.2013.10.024
Page 3, line 120-121 : brackets open but never closed.
Page 4, line 163-164: there is no information on how long the first and second stimuli presented, and the previous study was done for horses; readers benefit from explaining it here. Rocha’s stimulus exposure time was 5 min here 2 min.
Page 5, line 169: anti-clockwise according to the graph?
Page 5, line 184-185: The experiment could have been done better, alternating 6 cm vs 1 cm randomly. This way, authors may induce biased constantly exposing 6 cm slot first.
Page 5, line 193-197: Only one stimulus used and repeated morning and afternoon (pseudo-replication issue); what are the various time intervals? This makes it difficult to make comparison unless authors use rates.
Page 6, line 219-220: please cite the R software with the suggested standard reference format. The information available on the website and the version authors used.
Page 6, line 223 -224: I struggled to understand throughout the manuscript whether authors used match pair design: for example, matched the same individual between tests would have been the best way to compare inter-individual variations. Please see the above comments.
Page 8, line 270: frequency or total??
Page 8, line 274: where is figure 5??
Page 9, line 306: This is a problematic comparison. See major comments.
Page 9, line 308: do you have data to support the statement?
Author Response
Reviewer 1
Major comments
1
- The comparisons made between tests with the number of gazes and the number of scans invalid due to time slot differences. For example, the authors did not use the same amount of stimulus exposure time for visual and non-visual tests for count comparisons (VAT 2 min per individual, SVAT 8 min per individual). The gaze or scan rate would have been the best measure when the time slots were different in each test. As it stands, authors compared duration for two or, in some cases, three tests with varying times of exposure using counts as the response variable, and these comparisons were not valid. Authors could have used mixed models with Poisson or Negative binomial distribution for count response data to control the individual difference and the exposure time as the offset value to account for the differences.
Actually, since we were interested in individual variations WITHIN tests, we precisely did not do any comparison between tests in terms of number of glances and gazes, therefore there is no problem with these differences in time. Nevertheless, we have added figures with the number of gazes and glances per minute in response to the reviewer’s request. Line 169-176
Normality and homogeneity of variance were assessed by inspection of residuals with Shapiro-Wilk W tests. Given the non-normality distribution of data, we ran non-parametric statistical tests [50] using R 3.3.2 software® [51] with a significance threshold set at 0.05. The aim of this work was to assess individual variations in attention structure for each given test separately, using two aspects (number of glances and gazes, duration of gazes). Since the question and data were quite simple and the sample size reasonable, we stayed on “conventional” simple non-parametric statistical analyses. In the same line, we compared temporal changes in behavior within tests using Wilcoxon signed-rank test. Spearman’s correlations tests were used to assess possible individual consistency between tests (i.e. VAT; SVAT and AAT), since it compares the relative ranking of given individuals independently of the actual precise measures. In the figures, we have however used the rate per minute, in order to facilitate an overview of the results at the study’s level.
2
In the non-social visual object test (VAT), the time exposed the unusual object displayed on the screen not mentioned. The object presented with two different patterns (anti-clockwise as the figure indicated eight shapes). There was no information on how long each circulation pattern exposed to an individual within 2 min. I see authors refer to Rocha’s et al. 2017 for details, but again, the test was done for horses using different experimental setups, and the exposure time was different (5 min).
Yes thanks, we have added this information. Also we agree the information was not clear enough: actually the object was continuously moving, therefore it was slowly transforming into successive patterns following a circular movement. Song birds are known for presenting rapid shifts of attention, much more rapid than horses. This explains why we changed the duration for starlings (explained lines 118-124)
It was projected on the black screen during 2 minutes when the bird was positioned on the perch nearest the screen and facing it. The trajectory of the spot followed the same pattern of movement as in Rochais et al. [13] ‘s VAT, by using the Processing software©: the object was continuously moving, therefore it was slowly transforming into successive patterns following a circular movement (Figure 1). As suggested by Davidson et al 2014) [42], we adjusted the test duration to species-specific characteristics. Song birds are known for presenting rapid shifts of attention [43] much more rapid than horses. Each bird was tested four times, once each half-day for two consecutive days. Recording began 2 minutes before the stimulation was switched on and continued for 2 minutes during the stimulation. Thus, in all, we obtained 16 minutes of recording for each bird.
“Figure 1: Visual stimulus used during the non-social visual attention test, adapted from [26] The light was moved repeatedly, regularly and continuously over 20 seconds using a computer program, starting anticlockwise (a) to continue over a “8” shape (b). Six cycles were broadcasted during each 2 minutes test. Lines 126-128
In the social-visual test (SVAT), the conspecifics were always exposed with 6 cm slot first and then 1cm opening; clearly, an order effect influenced the results. The proper way to experiment to eliminate the possible order effect is randomly alternating the slots ( 6 cm or 1cm) per individual.
Pilot experiments had shown that the birds did not always see that there was a conspecific when the slot was small. Since we wanted to make sure the animals had all seen that they had access to a visible neighbor, we were obliged to have a fixed order, even though of course random allocation would have been ideal (lines 139-146)
The test bird (“A”) had visual access to his conspecific (“B”) twice a day for 5 minutes: in the morning, with the slot fully opened (6cm) so that bird “A” could learn the presence of his conspecific “B”; and in the afternoon, the slot was only opened 1cm to promote potential monocular observation of “B” by “A” (hence potentially laterality). We decided to systematically test our birds first in the morning with a large opening (6cm) for giving the possibility to the tested bird (A) to be aware of its congener presence (B) and to see him binocularly. In the afternoon, the limited opening (1 cm) allowed only monocular attentional behaviours. Pilot experiments had shown that the birds did not always see that there was a conspecific when the slot was small (see Range et al [19] who allow a familiarization period to birds before tests). Since we wanted to make sure the animals had all seen that they had access to a visible neighbor, we were obliged to have a fixed order, even though of course random allocation would have been ideal.
In the auditory test (AAT), the conspecific and heterospecific calls were obtained from a single individual respectively (page 5, line193-198), and playbacks prepared with replicating them. This playback experiment, unfortunately, a pseudo- replicated one and may not provide a valid test. Please see: Kroodsma et al .2001 (Animal Behaviour) DOI:10.1006/anbe.2000.1676
Again, thanks for your remark. We realized that we had not explained enough our procedure. There were several stimuli for each type of vocalization (hetero- and conspecific). We have added an explanation in the text.: lines 151-157
Two types of stimuli were broadcasted: (a) 4 different whistles from unfamiliar individuals recorded in New Zealand to ensure that the subjects could not have heard them before, and (b) 4 different vocalizations of humpback whales (Megaptera novaeangliae) as unknown heterospecific stimuli.. Durations (0.50sec) and intensities (~70dB at two meters) were standardized for all stimuli using ANA software [46]. Four different vocalizations (with different acoustic structures) of each type, one of each per half day, were broadcasted in random order and various time intervals (between 6 to 21 minutes) between stimulations to avoid biases due to habituation. Therefore, birds heard each stimulus only once. Video recording began 1 minute before the stimulation and finished 1 minute after the stimulation, in all we obtained 8 minutes of recording for each type of stimulus.
3
Throughout the manuscript, I struggled to understand whether the authors used matched-pair design to compare individuals between tests, despite the stimulus exposure time difference. For example, they matched the same individual between tests and reported the result or used the cumulative measure for all the individuals and compared the measures. This needs to be clarified, specifically where the correlation analysis performed (e.g., figure 4 a) and whether each data point represents the same individual tested in both tests (conspecific visual social vs conspecific vocalization).
As mentioned above, our aim was to characterize individual differences within tests, and thus no comparison of numbers was made between tests. The only time tests were compared was to assess individual consistency, by looking at possible correlations between tests. Since the Spearman test compares the ranks of the individuals, the absolute numbers are not important, only the respective ranking of each individual in each test. We tried to explain it more lines 221-224
Figure 3: a) Numbers of glances towards a conspecific during the visual social test (minimum opening) in relation to numbers of glances after the broadcast of a conspecific vocalization (Correlation rs=0.70, N=10, p=0.05). b) Total gaze durations after the broadcast of the conspecific vocalization in relation to total gaze durations following the hetero-specific vocalization during auditory tests (Correlation r=0.79, N=10, p. Each dot represents an individual.
Overall, the presentation of data can further improve. First, include the missing graph 5; then graph 4-a, is invalid as the time slot differences in stimulus exposure time. Figure 3 b, c showed redundant information and present in table 1 and 2; therefore, authors may include two graphs: 1) the number of glances and glance duration; 2) the number of gazes and gaze duration. Figure 1 c appeared in another published paper, indicated that authors use the same photo for several publications.
This was a mistake, the Figure 5 is now Fig 3. The figure 3 is valid, as explained above, this is a correlation, therefore not depending on absolute numbers. We have suppressed Figure 1 as requested and replaced by the reference to the paper line 150.
Moreover, in order to avoid redundancy but nevertheless provide useful information, we have put the tables in appendix (SM1 and SM2).
Minor comments
Page 1, line 31-32: The statement is invalid if the authors have used time interval for two tests; there is no information on how the green square move or the exposure time.
We explained better the protocol used in the non-social visual test and precisely how the green dot moves on the screen: line 118-124. See response above
Page 1, line 33: The individual variation between tests were not established; please see the major comments above.
This was not our primary aim. See response above.
Davidson et al., 2014. Animal Behaviour 87, 3-15 should be cited to give readers different gaze definitions. https://doi.org/10.1016/j.anbehav.2013.10.024
Thanks for the reference, we cited this reference lines 121-122 and line 255-257
As suggested by Davidson et al (2014) [42], we adjusted the test duration to species-specific characteristics.
The attentional structure of our starlings appeared to depend both on the targets and the modality involved. As Davidson et al (2014) [42]pointed out, gaze behaviours has to be described in term of “observable behaviour”. For example, a shorter latency and a longer time of looking may reflect gaze preference.
Page 3, line 120-121 : brackets open but never closed :
We closed the brackets
Page 4, line 163-164: there is no information on how long the first and second stimuli presented, and the previous study was done for horses; readers benefit from explaining it here. Rocha’s stimulus exposure time was 5 min here 2 min.
We explained lines 126-128 in the Figure 1 legend the duration on the stimulus and the reason of the differences between Rochais and the present study. See response above
Page 5, line 169: anti-clockwise according to the graph?
We have improved the description of the stimulus lines 126-128. See response above
Page 5, line 184-185: The experiment could have been done better, alternating 6 cm vs 1 cm randomly. This way, authors may induce biased constantly exposing 6 cm slot first.
We explain lines 139-146 why we presented first an opening of 6 cm and then an opening of 1 cm. See also response above
Page 5, line 193-197: Only one stimulus used and repeated morning and afternoon (pseudo-replication issue); what are the various time intervals? This makes it difficult to make comparison unless authors use rates.
We explain now lines 125-158 that the stimulus used for auditory test was different each time as there were 4 different starlings’ whistles and 4 different whales vocalizations. See also response above
Page 6, line 219-220: please cite the R software with the suggested standard reference format. The information available on the website and the version authors used.
We cited R soflware with the standard reference: line 434.
R Development Core Team. R: A language and environment for statistical computing. 2014; 07.10
Page 6, line 223 -224: I struggled to understand throughout the manuscript whether authors used match pair design: for example, matched the same individual between tests would have been the best way to compare inter-individual variations. Please see the above comments.
As responded above, we did not compare individual variations between tests apart from correlations, which are based on rankings.
Page 8, line 270: frequency or total??
We indicated that birds emitted more glances in total towards the loudspeaker: line 212-213
Overall, our subjects emitted more glances (total number) towards the loudspeaker in response to the heterospecific than to the conspecific stimuli (Wilcoxon signed-rank test, N=10, T=3, p=0.01)
Page 8, line 274: where is figure 5??
This was a mistake, sorry again, it is now Figure 2c
Page 9, line 306: This is a problematic comparison. See major comments.
We did not compare test directly as we mentioned above
Page 9, line 308: do you have data to support the statement?
Yes, we presented results in the Social visual Attention test section line 199-202
Interestingly, when the slot was fully open, although they had the possibility to use either their binocular or monocular visual fields to look at their conspecifics, starlings performed more monocular than binocular glances and gazes (TGlance=8, N=10, p=0.04; TGaze=8, p0.04). When using monocular vision, they still performed more glance than gazes (T=0, N=10, p=.005). However, we could not evidence a laterality bias (preferred side) (TGlance=17.5, N=10, p=0.3; TGaze=9.5, N=9p=0.1; TTotal gaze duration=4, N=7 p=0.09).

Reviewer 2 Report
The purpose of the study was to attempt to develop spontaneous attention tests with the aim to facilitate future research addressing attentional functions in songbirds within a comparative perspective. Three tests addressing diverse attentional aspects in different sensory modalities were adapted based on previous literature dealing with different species. Statistical analyses were focused on two measures of explicit attentional orienting, i.e. glances (short-lasting looks) and gazes (long-lasting looks). Total gaze duration towards the stimulus was also examined. The results showed that the different tests were able to capture different aspects of attention processes in starlings.
The experiments reported in the present paper seem to be characterized by a solid design and the paper is generally well-written. The topic is interesting and the findings may contribute to enrich current knowledge about attentional processing in songbirds and may stimulate further comparative investigations. I have only few, though relevant, concerns. These are related to the need to 1) deepen the analysis of the literature presented in the introduction/discussion sections, and 2) clarify and justify some methodological and statistical aspects.
1) In my view, both the introduction and the discussion sections would need a more focused (and updated) presentation of recent perspectives on the different attentional aspects dealt with by each different test administered in the study. This is particularly true for social visual attention. In this regard, two review papers may be particularly helpful:
Shepherd, S. V. (2010). Following gaze: gaze-following behavior as a window into social cognition. Frontiers in Integrative Neuroscience, 4, 5. https://doi.org/10.3389/fnint.2010.00005
Dalmaso, M., Castelli, L., & Galfano, G. (2020). Social modulators of gaze-mediated orienting of attention: A review. Psychonomic Bulletin & Review, 27, 833-855. https://doi.org/10.3758/s13423-020-01730-x
2) Sample size should be justified. The authors should also motivate the reason underlying the specific task order employed. It would also be interesting to know whether there were any rank differences in the pairs of individuals tested in the SVAT? This is potentially relevant because it is known that social status/dominance is positively correlated to cognitive ability in starlings (Boogert et al., 2006, Anim. Behav.), and it has been shown that some species show different (social) attention engagement depending on the status of the conspecific presented as stimulus (e.g., Dalmaso et al., 2012, Biol. Lett.; Shepherd et al., 2006, Curr Biol.). Finally, as concerns statistics, exact p values should always be reported throughout the results section (i.e., the authors should avoid expressions such as “p<0.02” or “p>0.05”) as well as in the tables.
As a final note, resolution of Figures 3 and 4 is unexpectedly poor. This needs to be fixed.
Other comments
- line 109. It seems that a parenthesis should be deleted.
- line 180. I would use "hereafter" rather than "here"
- line 216. "Belin et al 2018" is likely to be a typo and should be deleted.
- lines 292-293. The sentence is not clear and should be rewritten.
Author Response
Dear Editor
Please find a revised version of our manuscript “animals-1255122” We have thoroughly gone through both reviewers’ comments and made the modifications as best we could.
We hope this new version will fulfill the requirements to be published in this special issue of Animals.
Yours sincerely
Laurence Henry
Reviewer 2
1) In my view, both the introduction and the discussion sections would need a more focused (and updated) presentation of recent perspectives on the different attentional aspects dealt with by each different test administered in the study. This is particularly true for social visual attention. In this regard, two review papers may be particularly helpful:
Shepherd, S. V. (2010). Following gaze: gaze-following behavior as a window into social cognition. Frontiers in Integrative Neuroscience, 4, 5. https://doi.org/10.3389/fnint.2010.00005
Dalmaso, M., Castelli, L., & Galfano, G. (2020). Social modulators of gaze-mediated orienting of attention: A review. Psychonomic Bulletin & Review, 27, 833-855. https://doi.org/10.3758/s13423-020-01730-x
These references were added in the introduction and the discussion section : line 39, line 42, line 66, 285.
2) Sample size should be justified. The authors should also motivate the reason underlying the specific task order employed. It would also be interesting to know whether there were any rank differences in the pairs of individuals tested in the SVAT? This is potentially relevant because it is known that social status/dominance is positively correlated to cognitive ability in starlings (Boogert et al., 2006, Anim. Behav.), and it has been shown that some species show different (social) attention engagement depending on the status of the conspecific presented as stimulus (e.g., Dalmaso et al., 2012, Biol. Lett.; Shepherd et al., 2006, Curr Biol.). Finally, as concerns statistics, exact p values should always be reported throughout the results section (i.e., the authors should avoid expressions such as “p<0.02” or “p>0.05”) as well as in the tables.
The choice of sample size was based on experimental logistic constraints but also and mainly on a wish to keep a balance between statistical validity and the “reduction” principle of the 3R ethical principles especially as we used wild-caught animals. We have added this information lines 92_94.
We explained the order of the test line 104-108:
We chose to perform the non-social tests first as we supposed that the social tests may have a more excitatory or motivational effect on the birds which could then bias the results of the non-social tests (e.g. seeking more a conspecific behind the screen or the loudspeaker. The order of the the non-social tasks was changed between two subgroups of birds: Five birds were tested first in the non social-visual task and then in the auditory task and the other five birds were tested in the opposite order. During the second week the 10 birds were tested in the social visual test.
Pairs were randomly constituted and we do not take into account the possible hierarchical order between birds. Agonitic behaviour are quite limited in stable group and we did not test birds in dyadic confrontation before the 3 experiments (lines 1230-132)
For this test, five pairs of starlings were randomly constituted from the initial group of 10 subjects: observations in their aviary had revealed no clear hierarchical ranking, therefore, we could not take this parameter into account, although some authors found a relationship between hierarchical status and cognitive abilities in starlings [44] or between dominance and attention in other species [45].
And lines 283-295 integrated Dalmaso et al 2012 and Boogert et al 2006 references.
In addition, visual social attention was expressed mainly by glances in our experimental situations. Social life requires to be attentive towards others and their actions [21]. Attention and particularly social attention is modulated by social factors in most social species including human (Dalmaso et al (2012) [9, 45]. One important aspect is the quality of social relations between individuals such as hierarchical rank [64] or affinities ([30]. Blois-Heulin [18] and Lemasson et al. (2006) [65] showed opposite trends of visual attention between red-capped mangabeys (Cercocebus torquatus torquatus) characterized by a “despotic” type of society, where group members focus their attention towards the dominant male, and Campbell monkeys, characterized by a tolerant type of society, where members’visual attention network reflects affinities. In rhesus macaque (Macaca Rhesus) the lower ranking individuals reacted quicker than high ranking animals to visual cues from conspecifics [64]. Studies performed in large naturalistic aviaries have shown that most social relationships were expressed through tolerance and spatial proximities between preferred partners, while the hierarchy is more rarely expressed and only in some particular contexts [44, 66, 67]. Thus was also the case in the observations performed on our experimental birds when in group (Pougnault et al. in prep). In canaries, observational learning was enhanced when watching a preferred social partner [68]. We did not investigate the role of social status nor of being or not social partners on the attentional pattern. Further studies will have to be developed in order to test specifically this question. At least, these standardized tests can help testing further such questions.
We indicated the exact p-values throughout the results section
As a final note, resolution of Figures 3 and 4 is unexpectedly poor. This needs to be fixed.
We improved the quality of Figure 3 and Figure 4, now, figures 2 and 3.
line 109. It seems that a parenthesis should be deleted:
We closed by adding the missing parenthesis: “
line 180. I would use "hereafter" rather than "here" :
We change “here” for “herafter”
line 216. "Belin et al 2018" is likely to be a typo and should be deleted:
We deleted Belin et al 2018":
- lines 292-293. The sentence is not clear and should be rewritten.
These lines are now lines 229-230 and have been rewritten
However, in the auditory test, total gaze durations towards the loudspeaker after the broadcast of a conspecific song and after a hetero-specific song were positively correlated (Spearman’s correlation, rho=0.79, p=0.01; Figure 3b).

Round 2
Reviewer 1 Report
No further comments for the revised manuscript